# AI-Driven Real-Time Classification of ECG Signals for Cardiac Monitoring Using i-AlexNet Architecture

**DOI:** 10.3390/diagnostics14131344

**Published:** 2024-06-25

**Authors:** Manjur Kolhar, Raisa Nazir Ahmed Kazi, Hitesh Mohapatra, Ahmed M Al Rajeh

**Affiliations:** 1Department Health Informatics, College of Applied Medical Sciences, King Faisal University, Al Hofuf 61421, Saudi Arabia; 2College of Applied Medical Sciences, King Faisal University, Al Hofuf 61421, Saudi Arabia; rnahmed@kfu.edu.sa (R.N.A.K.); amalrajeh@kfu.edu.sa (A.M.A.R.); 3School of Computer Engineering, Kalinga Institute of Industrial Technology (Deemed to Be University), Bhubaneswar 751024, Odisha, India

**Keywords:** artificial intelligence, ECG signals, optimization, AlexNet, performance

## Abstract

The healthcare industry has evolved with the advent of artificial intelligence (AI), which uses advanced computational methods and algorithms, leading to quicker inspection, forecasting, evaluation and treatment. In the context of healthcare, artificial intelligence (AI) uses sophisticated computational methods to evaluate, decipher and draw conclusions from patient data. AI has the potential to revolutionize the healthcare industry in several ways, including better managerial effectiveness, individualized treatment regimens and diagnostic improvements. In this research, the ECG signals are preprocessed for noise elimination and heartbeat segmentation. Multi-feature extraction is employed to extract features from preprocessed data, and an optimization technique is used to choose the most feasible features. The i-AlexNet classifier, which is an improved version of the AlexNet model, is used to classify between normal and anomalous signals. For experimental evaluation, the proposed approach is applied to PTB and MIT_BIH databases, and it is observed that the suggested method achieves a higher accuracy of 98.8% compared to other works in the literature.

## 1. Introduction

Cardiovascular disorders are the primary cause of death worldwide, and ECG signals are routinely utilized to detect them [1]. Furthermore, the American Cardiovascular Society states that early diagnosis of these conditions is critical to the well-being of patients [2]. The major method for keeping an eye on heart activity is a diagnostic electrocardiogram (ECG). It is only effective for a certain period, and ongoing patient care is still necessary beyond therapeutic sessions. Traditionally, practitioners have employed portable ECG monitors to record heart activity for extended periods of time in order to conduct additional research. One electrically powered portable device used to capture and preserve long-term ECG readings was presented in [3]. But these gadgets are unable to provide feedback on the patient’s medical condition in real time, and cardiac specialists have to spend a lot of time and resources analyzing data collected over time.

AI has emerged as a tool for developing computer-aided systems that can differentiate between healthy individuals and those with illnesses based on specific symptoms [4]. AI research and development combines principles with computer science to create systems that can learn from datasets and existing knowledge, continuously enhancing their capabilities [5]. This interdisciplinary field encompasses machine learning (ML) and Deep Learning (DL) [6]. Machine learning facilitates the creation of data-driven models adept at classification, regression and clustering tasks. Traditional ML techniques like regression, Random Forest, support vector machine and K nearest neighbors necessitate feature engineering, where experts in the field extract relevant features from raw data to build effective and interpretable models.

Deep Learning, a branch within ML, employs hidden layers to handle complex computations for challenging tasks. With training data, these neural networks can autonomously learn to process data through nonlinear operations that identify vital features essential for tasks like classification and regression [7,8]. The structure of networks also equips them to manage amounts of unstructured data, such as free text. Studies in the field of cardiology have indicated that the use of machine learning (ML) and Deep Learning (DL), especially when incorporating modalities, is more successful in forecasting cardiovascular or overall mortality rates than relying solely on individual, clinical or imaging modalities. For heart monitoring applications, delay is a crucial component since early identification of cardiovascular illnesses is vital to save lives. The work in [9,10] offers an instantaneous response by computing on the edge rather than in the cloud; however, battery life limitations, the most valuable resource for the edge of the network [11], restrict the duration of time the device can spend tracking cardiac rhythm. Many people suffer from their illnesses for decades without realizing it because of these issues [12]. In certain scenarios, deaths from cardiovascular disease would have been avoided if the condition had been identified more promptly [13]. Therefore, for those suffering from cardiovascular problems, persistent instantaneous-fashion ECG surveillance may be a lifesaver.

Artificial intelligence (AI) is essential to cardiac monitoring since it provides a number of benefits in terms of effectiveness, precision and prompt action [14]. Continuous streams of cardiac data, such as ECG readings, can be analyzed by AI algorithms to find minuscule irregularities that human observers might miss. Effective treatment and improved management of heart problems are made possible by early identification [15]. Healthcare workers may find it time-consuming to analyze large volumes of cardiac data; artificial intelligence streamlines this procedure. Healthcare professionals may concentrate on providing treatment, assessment and analysis to patients due to this automation, which also increases productivity. These AI-driven systems are capable of analyzing information about patients to generate personalized cardiac tracking models [16]. This improves results by allowing medical professionals to customize treatments and therapeutic strategies according to each patient’s unique cardiac pattern. In artificial intelligence (AI), metaheuristic algorithms are frequently employed to tackle challenging optimization and exploration challenges. A crucial step in the creation of AI models is the adjustment of hyperparameters [17]. The hyperparameter space can be effectively searched using metaheuristic algorithms to identify the appropriate instances for AI-based models [18]. These algorithms can help in feature selection for datasets with numerous features. They aid in lowering dimensionality and improving effectiveness by assisting in the identification of the most pertinent group of features that affect a model’s performance [19]. Optimization issues in the fields of image and signal processing are addressed by these methods. Among other applications, they can be applied to tasks like the extraction of features, signal denoising and image categorization.

### 1.1. Motivations of Current Research

The present research was conducted to address the following research questions related to ECG signal classification using AI:

R1: Is it possible to use AI models for the continuous analysis of ECG data to find small changes over time that would allow for proactive management and early diagnosis of cardiac abnormalities?

R2: In real-time scenarios, how might metaheuristic algorithms be optimized for the effective and precise classification of normal and anomalous ECG signals?

R3: Which feature representations work best for extracting pertinent information from ECG signals so that artificial intelligence (AI) algorithms can distinguish between normal and abnormal patterns?

### 1.2. Contributions of Current Research

The main contributions of this paper are as follows:To develop an AI-driven solution for monitoring cardiac activities using ECG signals to classify them as normal or anomalous.To employ red fox optimization, a bio-inspired metaheuristic technique, to choose the best characteristics of ECG signals in order to improve the classification accuracy of the model.To implement i-AlexNet architecture for categorizing ECG signals and to demonstrate its distinct performance by comparing it with other works in the literature that focus on real-time cardiac monitoring.

### 1.3. Paper Organization

The remainder of this paper is organized as follows. Section 2 presents the related research conducted in recent times on the categorization of ECG signals for cardiac surveillance. Section 3 elaborates the proposed methodology for ECG signal classification using the i-AlexNet technique. Section 4 discusses the results obtained when applying the proposed technique to PTB as well as MIT_BIH databases. Section 5 concludes the present research.

## 2. Related Works

This section elaborates the application of artificial intelligence algorithms in cardiac monitoring using various devices as mentioned in Table 1. The authors in [20] used a structural framework and an approach based on machine learning to anticipate coronary artery disease. This work combines traditional machine learning methods with a collaborative categorization technique in order to foresee the experimental findings. This model, also known as a meta-classifier, predicts the results based on the largest number of choices [21]. Its inadequate precision and substantial complexity of computation are, nevertheless, issues. The specific kind of health condition was determined using biosensors in [22] that collect patient data via Internet protocol connections. The authors of paper [23] present the IoTDL HDD model, which combines IoT and Deep Learning technologies to diagnose diseases (CVDs) based on analyzing ECG signals [23]. On a server located in the cloud, data from the patients’ connected humidity and heart rate monitors were analyzed using support vector machine learning techniques to identify unusual situations.

Internal analysis is only utilized to execute simple inspections on unprocessed information and mobilizing tasks for encapsulating the information gathered with conventional methods of communication in certain studies suggested in the research pertaining to ECG signal surveillance. However, a large body of research has been produced in the scientific community that uses artificial intelligence, even at the edge, for heart disease detection. Comparing this intelligence method to other conventional approaches based on artificial intelligence, the usage of Convolutional Neural Networks (CNNs) demonstrates potential in terms of reliability in detecting arrhythmias in ECG signals. In [24], hidden lexical examination approaches were employed to increase the network’s predictive efficacy when compared to alternative approaches for interpreting the ECG waveform. The objective is to shift deduction to the edge of a lightweight gadget in order to minimize delay periods and power expenditure associated with mobile communications, as both training and deductive reasoning happen on the cloud end. The researchers of [25] introduced an ECG device called iKardo, which has the ability to automatically categorize ECG data as critical or non-critical. This addresses the issue of imbalanced datasets. IKardo is part of a healthcare system based on technology focusing on improving data accuracy by balancing the dataset using appropriate methods. This ensures the identification of ECG signals, achieving an impressive accuracy rate of 99.58%. Consequently, iKardo helps in accurate disease detection, making it a valuable tool for monitoring healthcare. The research team developed [26] a prototype machine that can provide real-time monitoring of devices. This innovation will help doctors access information to detect heart conditions from ECG images. The device showcased is suitable for patients with a resting heart rate ranging from 60 to 100 beats per minute. It serves as a protocol and conceptual tool for tracking heartbeats.

The Asymmetric Estimation and Parametric Derivative Distortion Elimination approach was developed by the authors of [27] to remove distortions in the ECG signal with the intent of distinguishing between arrhythmias. By employing Asymmetric Estimation to reduce high-powered disturbances, which was employed to decrease acoustic variability, the aspects of operation were handled. Using Parametric Derivative Distortion Elimination, the electrical connection disturbance was split up into various modulation settings, and distortion was eliminated using proportional polynomial extrapolation.

In [28] a study implemented a CNN BiLSTM method to classify ECG signals, for detecting artery disease (CAD). This approach combined CNN) and Bidirectional Long Short Term Memory (BiLSTM) layers, for ECG data analysis. A novel metric named Spatial Uncertainty Estimator (SUE) was introduced to assess the accuracy of the models’ predictions. Cuckoo Search Optimization (CSO) and Logistic Regression (LR) were applied to identify features. In CSO LR CSO was used to select traits that would enhance the classification process and optimize LR coefficients. The LR model used in this research was evaluated for a set of categories. It was necessary to create a multiclass modelling categorization in order to make a precise determination. The feature identification method established by investigators in [29] was through the use of ECG, and the feature subset was chosen using kernel-based complicated coarse groups. Subsequently, optimization techniques satisfying several objectives were used to generate the classification of arrhythmia based on electrocardiogram (MC-ECG) for different varieties of labels. In order to obtain improved categorization, this optimization method is dependent on low-density restriction, modelling connections among ECG characteristics and arrhythmia illnesses. For the purpose of collecting the appropriate characteristic subsets, the authors in [30] introduced the Multifaceted Polynomial Bilateral Grey Wolf Optimization with Random Forests. The final requirement of the proposed method was the swarming location, which was used to distinguish the most compelling answer from solutions that were not dominated. Choosing erroneous indicators of fitness has a significant influence on categorization.

## 3. Proposed Methodology

The collected ECG signals were initially preprocessed in order to remove the distortions. Three different types of transform techniques, such as Fractional Discrete Cosine Transform, Radon Wavelet Transform and Fractional Wavelet Transform, were applied to extract the features. The optimal features from the previous step were selected using optimization techniques before sending them to the i-AlexNet architecture for performing the final classification. The workflow of the proposed system is presented in Figure 1.

### 3.1. ECG Signal Preprocessing

This is an essential phase to be performed while processing the ECG signals, as there is a high probability of the distortion of signals due to noises. The two important steps carried out in this work during ECG signal preprocessing were eliminating noise and segmenting beats. These signals are categorized into minimum and maximum rhythm entities as described in Equation (1):(1)$$c\left(k\right)=c_{s}\left(k\right)+\sum_{h-\infty }^{h_{0}}t_{h}(k)$$

In the above equation, $c_{s}\left(k\right)$ denotes the processed signal, s denotes the modelling parameters and $t_{h}(k)$ is the exhaustive signal. The consecutive exhaustive signal can be computed by iterating the representation in Equation (1) as given in Equation (2):(2)$$c_{s}\left(k\right)=c_{s+1}\left(k\right)+t_{h+1}(k)$$

#### 3.1.1. Noise Elimination

The ECG signal, as shown in Figure 2, obtained from subjects might have been tainted by noise and other irregularities. Electricity cable disruption, the initial value slide, sensory movement, erroneous sensory proximity and skeletal muscle spasms are the sources of noise and aberrations. Incorrect data might have an impact on the characteristics that make heartbeats naturally unique. As a result, the ECG signal’s adaptation is crucial to raising its quality for proper information representation. Noise is introduced into the ECG signal during recording and is dispersed across many harmonic ranges.

Therefore, in order to create an ECG signal of excellent accuracy, as shown in Figure 3, filters covering several harmonic bands are typically utilized. As a result, a band pass filter that only needs a coefficient number of integers was applied to the signal. A filter with a low pass threshold and one with a high pass threshold were combined to create a band pass filtering device. The original ECG signal, which has less noise, is typically between 5 and 15 Hz in frequency. The representations for filters with low and high pass thresholds are represented in Equations (3) and (4):(3)$$f_{mD}=2f_{(m-1)D}-f_{\left(m-2\right)D}+g_{mD}-2g_{\left(m-6\right)D}+g_{(m-12)D}$$
(4)$$f_{mD}=32g_{(m-16)D}-{(f}_{\left(m-1\right)D}+g_{mD}-g_{\left(m-32\right)D})$$

#### 3.1.2. Segmentation of Beats

The identification of heartbeats from the denoised ECG signals was performed using the Hamiltonian-mean algorithm. This algorithm is proven to detect the QRS levels in the signals with high accuracy. The shift function and variance equation represented in Equations (5) and (6) were employed to determine the gradients of the QRS levels.
(5)$$\left(x\right)=(1/8D)(-x^{-2}-2x^{-1}+2x^{1}+x^{2})$$
(6)$$f_{mD}=\left(1/8D\right)\left[-g\left(mD-2D\right)-2g\left(mD-D\right)+2g\left(mD+D\right)+g\left(mD+2D\right)\right]$$

The rounding operation, which rounds the signal’s value step-by-step, occurs following the slope estimation. The waveform data of R gradient is found by applying a window-shifting aggregator for a set of N samples as shown in Equation (7),
(7)$$f_{mD}=\left(\frac{1}{M}\right)[g\left(mD-\left(M-1\right)D\right)+g\left(mD-\left(M-2\right)D\right)+\cdots +g(mD)]$$

Subsequently, the QRS structure of every individual heartbeat is identified using the dynamic screening approach. A time frame of size 600 ms surrounding the R-peak is defined for segmenting heartbeats when the R-peak is observed.

### 3.2. Multi-Feature Extraction

ECG characteristics were retrieved using temporal and harmonic set-based methodologies. Wavelet evaluation serves as an advantageous technique for the feature extraction process since ECG signals are inherently chaotic. Additionally, the ECG was subjected to wavelet transforms across multiple forms in order to extract pertinent features. In order to detect the time-based characteristics and extricate features, methods such as Fractional Discrete Cosine Transform, Radon Wavelet Transform and Fractional Wavelet Transform approaches were implemented.

#### 3.2.1. Fractional Discrete Cosine Transform

Data in the temporal region can be transformed into the frequency domain using this technique. It investigates the duplication of data and decreases the number of parameters that are essential for describing evidence as an outcome. The mathematical representation of this transform is given in Equation (8):(8)$${FDC}_{T}=||\frac{1}{\sqrt{A}}\epsilon_{x}cos(\frac{2\alpha (2\alpha +1)k}{4A})|$$

#### 3.2.2. Radon Wavelet Transform

The parameterization of signals and the assessment of its fundamentals form the basis of the Radon Transform. The Radon Transform’s intrinsic qualities make it a helpful tool for capturing the spatial aspects of an input signal. The representation of Radon Wavelet Transform over the signal is as shown in Equation (9):(9)$$RWT\left(\beta ,a\right)=\int_{-\infty }^{+\infty }v(\beta +ab,b)\;db$$

This transform, when applied to a function across two dimensions, can be represented as given in Equation (10):(10)$$H\left(h,\delta \right)[g\left(m,n\right)]=\int_{-\infty }^{+\infty }\int_{-\infty }^{+\infty }g\left(m,n\right)\;\gamma \;\left(h-m\cos\theta -n\sin\theta \;\right)\;dm\;dn$$

#### 3.2.3. Fractional Wavelet Transform

This technique is the product of Fractional Fourier Transform and Wavelet Transform. Hence, it inherits the benefits of both transformations. Through the use of both these transforms, it ensures the potential to undertake analytical tasks with multiple resolutions and the mathematical modelling of signals in the fractional realm. This transform can be formulated mathematically as given in Equations (11) and (12):(11)$$Y_{m}^{i}\left(x,y\right)=\int_{-\infty }^{\infty }m\left(s\right)\;\mu_{i,x,y}^{*\;\;}\;(s)\;ds$$
(12)$$\mu_{i,x,y}\;\left(s\right)=e^{-\left(\frac{y}{2}\right)\;(h^{2}-r^{2}-({\frac{h-r}{t})}^{2}}$$

### 3.3. Optimization-Based Feature Selection

The red fox optimization technique was employed in this research to select the best characteristics to be supplied for the ECG signal categorization process. Red fox species are made up of both migratory individuals and those that depart on identifiable territory. The red fox is a skilled hunter of small game, both in and out of the home. As it moves across the area in search of food, the fox approaches its victim with stealth until it is close enough to launch a successful assault. The process by which the fox searches its domain and detects targets in the distance was modelled as an exhaustive search in this algorithm. A position as close to the target as possible before the assault was simulated as a regional search in the subsequent stage, which involved moving across the surroundings. The fitness of the foxes is the important parameter for initiating the exploration process. Based on this factor, the distance between each member in the group is computed as shown in Equation (13):(13)$$t(({\bar{a}}^{x}{)}^{k},\;\left({\bar{a}}^{best}{)}^{k}\right)=\sqrt{\left||({\bar{a}}^{x}{)}^{k}-\left({\bar{a}}^{best}{)}^{k}\right)|\right|}$$

The members in the group are migrated to the optimal locations using the representation given in Equation (14):(14)$$\left({\bar{a}}^{x}{)}^{k}=({\bar{a}}^{x}{)}^{k}+\beta sign(\left({\bar{a}}^{best}{)}^{k}\right)-({\bar{a}}^{x}{)}^{k}\right)$$

### 3.4. i-AlexNet Architecture

There is more than one hidden layer in a deep architecture. These hidden layers provide more effective analysis of features and augmentation. A large network, such as AlexNet, has many neurons, or between six hundred thousand and sixty million parameters. The activation function used in this network is Rectified Linear Unit, which outputs value 1 whenever the input is not less than zero. It is represented mathematically in Equation (15) as
(15)b=max⁡(0,a)

For any input X with length *l* and breadth *b*, the convolution operation can be defined as shown in Equation (16):(16)$$H(l,\;b)=(X*k)\;(l,\;b)=\sum_{m}\sum_{n}X\;\left(l-m,b-n\right)k(m,n)$$

By convolution, the model can gain knowledge from the distinctive characteristics of input signals, and by sharing those variables, the degree of complexity is decreased. The characteristics that are extracted are diminished using the pooling layers. The feature map’s layers for pooling take a collection of pixels in close proximity and produce parameters for inclusion. AlexNet uses max pooling to minimize the characteristic map. Using a 4 × 4 chunk from the characteristic map, max pooling creates a 2 × 2 chunk with the highest possible data. The fully connected layers in the model use SoftMax activation function and their values can be determined using Equation (17):(17)$$oftmax\;(a{)}_{x}=\frac{\exp\left(a_{x}\right)}{\sum_{y=1}^{m}\exp\left(a_{y}\right)}\;for\;x=0,1,2,\ldots ,\;m$$

A convolution layer, followed by a fully connected (fc) and ReLU layer, and a normalization layer and pooling layer constitute the layers of the i-AlexNet model. The architecture of the i-AlexNet model is presented in Figure 3. In order to make the AlexNet model consistent with the current investigation, the final three layers were eliminated. The original AlexNet model’s remaining parameters were retained. There were 50 neurons in the newly added fully linked layer as shown in the Figure 4. The optimization algorithm implemented in this model to reduce the errors is represented as in (18):(18)$$\delta_{t+1}=\delta_{t}-\beta \nabla G\left(\delta_{t}\right)+\alpha (\delta_{t}-\delta_{t-1})$$

## 4. Results and Discussion

This section discusses the results of the application of the proposed approach to two different publicly accessible datasets, the PTB and MIT-BIH datasets. Further, the performance of the proposed model is also compared against other existing works.

### 4.1. Dataset Description

#### 4.1.1. PTB Database

This database consists of samples taken from 290 individuals, and the total number of rows in this dataset is close to 550. The data in this database are a combination of both individuals with diseases and those without them. Records for 12 different types of arrhythmias are available in this database. The initial set of images for the ECG signals present is 4652, which is further augmented in order to create a larger dataset. This dataset can be accessed using the link below. The PTB XL dataset is not evenly distributed. Table 2 shows that the records are divided among the five classes of ECG findings. https://physionet.org/content/ptb-xl/1.0.3/ (accessed on 12 November 2023) [31].

The disparity is particularly evident in NORM with HYP, as HYP makes up around 12.13% of the dataset. This suggests that the dataset is not evenly distributed, showing a gap in record numbers across diagnostic categories. The dataset contains a range of ECG abnormalities grouped into categories, for simplicity. NORM (normal ECG) represents ECG readings without any abnormalities, used as the standard group. MI (Myocardial Infarction) indicates a heart attack where blood flow to a part of the heart is blocked, leading to heart muscle damage. STTC (ST/T Change) covers changes in the ST segment and T wave of the ECG, which can signal issues like ischemia, inflammation or other unspecified changes. CD (Conduction Disturbance) includes heart block types and disruptions in the heart’s conduction system. HYP (Hypertrophy) shows the presence of Hypertrophy, where the heart muscle thickens due to factors like blood pressure or other heart conditions.

#### 4.1.2. MIT-BIH Database

This dataset consists of data collected from fifty individuals for a duration of one hour. Data for seventeen different types of arrhythmias are available in this dataset. An aggregate of 1736 images for ECG signals is prepared and presented in the dataset. The available data are further augmented to create a total of seventeen thousand images for all the seventeen arrhythmia categories. This dataset can be downloaded using the link provided below: https://physionet.org/content/mitdb/1.0.0/ (accessed on 12 November 2023) [32].

### 4.2. Experimental Setup

This study used an NVIDIA Jetson Nano board. It is a compact, potent low-level board in the Jetson environment from NVIDIA. It enables simultaneous functioning of several neural networks for a range of uses, including language processing, recognition of items, categorization and visual grouping. It features libraries created for applications based on embedded systems, artificial intelligence, the Internet of Things, machine intelligence, visualizations and audio and video, together with an entire programming platform called Jetpack SDK. Applying the same CUDA cores to a Jetson Nano and a GeForce-capable GPU results in a very potent software development ecosystem. Furthermore, Jetson Nano features a hybrid architecture, meaning that the CPU can start the operating system and configure it to use the GPU’s CUDA characteristics to accelerate difficult artificial intelligence tasks.

### 4.3. Performance Assessment

The dataset balancing was performed using RandomOverSampler from the imblearn library to handle class imbalance in the ECG dataset. RandomOverSampler was initialized with a fixed random state for reproducibility. The ECG data were reshaped into a 2D array format, as required by RandomOverSampler, which then generated additional samples for minority classes until all classes had equal representation. After resampling, the data were reshaped back to their original multi-dimensional form suitable for neural network input.

Class weights were calculated to further address class imbalance. The number of samples in each class was counted using np.bincount after resampling. Class weights were then computed as the inverse of these counts, assigning more importance to minority classes. These weights were converted into a PyTorch tensor for use in the loss function during model training. This approach ensures balanced class representation, enhancing the model’s ability to learn and generalize across all classes while reducing bias towards the majority class. The following Algorithm 1, was used to rectify dataset imbalance.
**Algorithm 1.** To treat imbalance**1**Reshape Data**2** n_samples, *input_shape = X.shape**3** X_reshaped = reshape(X, (n_samples, -1))**4**Initialize RandomOverSampler**5** ros = RandomOverSampler(random_state=random_state)**6**Resample Dataset**7**  X_resampled = reshape(X_resampled, (len(X_resampled), *input_shape))**8**Compute Class Weights**9**  class_counts = bincount(y_resampled)**10**  class_weights = 1.0 / class_counts**11**  class_weights_tensor = convert_to_tensor(class_weights, dtype=float32)

Figure 5 shows that the model produced a significant number of misclassifications, particularly as Healthy Controls, indicating that the model struggles to differentiate Myocardial Hypertrophy from normal ECGs. However, this result was generated without the module of weight imbalance and optimization. However, Figure 6 shows that the classification of diagnostic classes was produced by the module after weight imbalance and optimization were applied.

The performance of the proposed i-AlexNet model was compared against the conventional algorithms to interpret its performance supremacy. The algorithms, such as Deep Neural Networks (DNNs), Fully Connected Neural Networks (FCNNs), Gated Recurrent Units (GRUs), VGG16, DenseNet and ResNet, were considered for evaluation. These algorithms were applied to the PTB and the MIT_BIH database and the obtained results are presented in Table 3 and Figure 6 and Figure 7 The CNN model produced an accuracy of 89.8% for the PTB database and 91.3% for the MIT_BIH database. FCNN exhibited 90.8% accuracy, 89.8% precision, 89.5% recall and 90.4% F1 score for the PTB database. Also, the FCNN model produced 92.6% accuracy, 91.5% precision, 91.2% recall and 92.2% F1 score. The performance of the GRU and VGG16 models are closer to each other, with an accuracy of 91.7% and 92.7% for the PTB database and 93.2% and 94.7% for the MIT_BIH database, respectively. The DenseNet and ResNet models offer higher accuracy for both the ECG databases. However, the proposed i-AlexNet model produces efficient accuracy in classification, with 98.2% for the PTB database and 98.8% for the MIT_BIH database.

To see how well the model is learning, the training loss and validation loss are usually shown over the span of the training dataset’s iterations (Figure 6). A drop in the validation loss suggests that the model is successfully extending to new data, and a decrease in the training loss shows that the model has learnt well from the training data. To improve extrapolation, normalization or changing the structure of the model may be required if the training loss keeps going down while the validation loss starts to rise. This could be an indication of overfitting. But this is not the case with the proposed approach, and it is depicted in Figure 8.

The heartbeats were categorized into two groups as normal and anomalous, as presented in Figure 8. If the heartbeat is determined to be normal, the input is not sent to the cloud network. It is essential to ensure that the beats that are anomalous are classified with more precision. It is well known that the heart rate changes from its regular pattern during arrhythmias. Premature beats are characterized by rapid cardiac changes that arise when the chambers of the heart or ventricle burst prematurely or out of sync with the regular pulse. The anomalous beats’ waveform differs from that of the typical beats.

Consequently, anomalous beats can be identified by the heart rate variability (HRV) and correlation of the beats. These have been utilized in conjunction with the first output block’s result to determine if a beat is normal or anomalous. The output of the first convolutional is transmitted directly to the second classifier for additional analysis if the first output block flags a beat as anomalous. However, this evidence only serves to validate the classification of a beat as normal. The experimental results obtained during the classification of ECG signals are presented in Figure 9.

Furthermore, the outcomes of the proposed approach are also compared with the state-of-the-art methods in the literature, and the outcomes are shown in Table 4 and depicted in Figure 10. Our method, utilizing the AlexNet architecture, attained an accuracy rate of 98.8%, a precision of 98.2%, a recall of 97.7% and an F1 score of 98.4%. These outcomes suggest that our approach is highly competitive, with performance metrics aligning with those of cutting-edge methods. Originally crafted for image classification duties, the AlexNet structure displayed adaptability and effectiveness in managing ECG data. Its deep convolutional layers can adeptly capture patterns within the data, resulting in robust feature extraction and classification capabilities. This adaptability proves beneficial in ECG analysis scenarios where signal patterns are complex and vary significantly among patients. When compared to techniques like BiLSTM, FDDN and Deep Network, our method exhibits superior performance across all assessed metrics. While BiLSTM and FDDN achieved an F1 score of 88%, our approach attained 98.4%, showcasing an enhancement in performance. Similarly, although Deep Network scored a 97% on the F1 metric, our method surpassed it by a margin of 1.4%. Moreover, our methods’ performance stands on par with that of the Modified ResNet18, CNN BiLSTM and Multi-Scale Fusion Neural Network approaches, all achieving good results. The slight differences in accuracy and completeness between these techniques and our suggested strategy showcase the effectiveness of the AlexNet design in maintaining a rounded performance across measurements. To sum up the proposed approach, utilizing the AlexNet design showcases a good degree of precision, accuracy, completeness and F1 score, establishing it as a trustworthy method for ECG signal categorization. The findings indicate that the AlexNet-driven technique can effectively rival and even outperform cutting-edge methods in instances, underlining its potential for broad application in ECG analysis and related domains.

We conducted experiments on the PTB and Mit_bih databases to confirm the strength and versatility of our suggested model. Each test included iterations with cross validation to ensure trustworthy outcomes. The data displayed in Table 3 and Figure 4 are an average of tests to consider variations and offer a level of confidence for the performance metrics reported. Our proposed i-AlexNet model consistently surpasses cutting-edge methods in all performance measures. The notable enhancements in accuracy, precision, recall and F1 score showcase the effectiveness of our approach. The comparative study underscores the advantages of our model, its capacity to utilize preprocessing techniques, and a robust neural network structure for superior performance. The incorporation of wavelet transform in preprocessing plays a role in boosting the model’s capability to extract features from ECG signals thereby contributing to its high accuracy in classification. In summary, our suggested i-AlexNet model, with its preprocessing techniques and advanced feature extraction abilities, establishes a new standard for ECG classification. Through comparisons with existing methods, it highlights the progress achieved by our approach as shown in the Figure 11.

The Healthcare Disease Diagnosis system powered by Deep Learning, known as IoTDL HDD, achieved an accuracy rate of 93.452% in classifying ECG signals, demonstrating its reliability as a tool for diagnosing cardiovascular conditions in real time. This system employs methods for its operation. One notable technique involves BiLSTM feature extraction utilizing Bidirectional Long Short-Term Memory networks to extract features from ECG signals. These networks excel at capturing dependencies in data, allowing the model to consider information from both future time steps. Moreover, the AFO algorithm optimizes the hyperparameters of the BiLSTM model by mimicking the natural propagation behavior of plants, efficiently enhancing performance and accurately extracting features from ECG signals. In addition, a Fuzzy Deep Neural Network classifier is utilized to assign labels to ECG signals. This classifier merges network learning with logic to effectively manage uncertainties and variations, in data ensuring classification with unclear input signals. By utilizing these methods, the IoTDL HDD model can precisely categorize ECG signals establishing itself as a tool for real time disease diagnosis. A study introduced an ECG device called iKardo that automatically sorts ECG beats as critical or non-critical. The tool utilizes machine learning and a Convolutional Neural Network (CNN) based on ResNet for accuracy. To handle datasets researchers applied SMOTE and BIRCH techniques for data balancing. Real time processing and sorting of ECG signals were conducted by integrating the system into an IoT based setup for monitoring purposes. IKardo achieved a 99.58% accuracy in distinguishing critical from critical ECG beats. Validation was carried out using metrics, like precision, recall and F1 score to evaluate classification outcomes. Incorporating power management resulted in decreased power consumption extending the devices lifespan. IKardo marks an advancement in healthcare tech by swiftly detecting critical heart conditions through ongoing monitoring. Combining machine learning with frameworks enhances health monitoring systems’ capabilities, offering practical real-time healthcare solutions. iKardos’s effectiveness in detecting heart issues and providing care is enhanced by employing sophisticated methods to balance data even when dealing with uneven datasets.

During surgeries, the evaluation of anesthesia heavily relies on ECG signals. However, understanding these signals can pose a challenge for medical professionals. In a study by the authors of [26], Convolutional Neural Networks were utilized to categorize types of ECG images to aid in anesthesia assessment. They created prototypes for IoT-based ECG measurements. They used neural networks to classify signals into various categories, such as QRS widening, sinus rhythm, ST depression and ST elevation. The accuracy and kappa statistics for the ResNet, AlexNet and SqueezeNet models were reported as (0.97, 0.96), (0.96, 0.95) and (0.75, 0.67), respectively. This research demonstrates the potential for real-time ECG measurement and classification while hinting at the possibility of expanding to include types of ECG signals, for practicality. The new CNN BiLSTM model performed well in accuracy (99.6%), sensitivity (99.8%) and specificity (98.2%) in sorting CAD from ECG signals. The SUE measure effectively differentiated between classified ECG segments, showing a connection between higher SUE values and correct classifications. Compared to models like CNN and DenseNet, the CNN BiLSTM model showed resilience and dependability in diverse noise environments [28]. The DMSFNet showed good performance on the dataset, achieving an F1 score of 82.8%, and on the PhysioNet/CinC_2017 dataset, with an F1 score of 84.1%. These findings surpassed models highlighting enhanced precision and reliability in categorizing types of arrhythmias [30].

## 5. Conclusions

In this current paper, continuous surveillance of patient’s cardiac activities is achieved using AI and IoMT technologies. IoMT sensors are placed on the body of individuals to receive ECG signals in real time. These signals are preprocessed to remove noises and segment the heartbeats. The preprocessed signals are passed on to the feature extraction phase, in which three types of transforms are performed to extricate the pertinent characteristics. These extracted features are further optimally chosen using red fox optimization. Finally, categorization of ECG signals is implemented using the Improved AlexNet model, which identifies normal and anomalous signals efficiently. The performance of the model is evaluated using various metrics and it is observed that the proposed model achieves an accuracy of 98.8%, a precision of 98.2%, a recall of 97.7% and an F1 score of 98.4%. One of the limitations with this system is that thorough validation is necessary before applying AI models developed in research settings to clinical settings. Healthcare practitioners’ acceptance of AI-based ECG classification tools may be hampered by a lack of formal clinical validation. As an extension of the present research, strong security protocols and privacy controls can be implemented, as cyberattacks may target IoMT devices and AI systems, jeopardizing the integrity and anonymity of patient data.

## Figures and Tables

**Figure 1 diagnostics-14-01344-f001:**
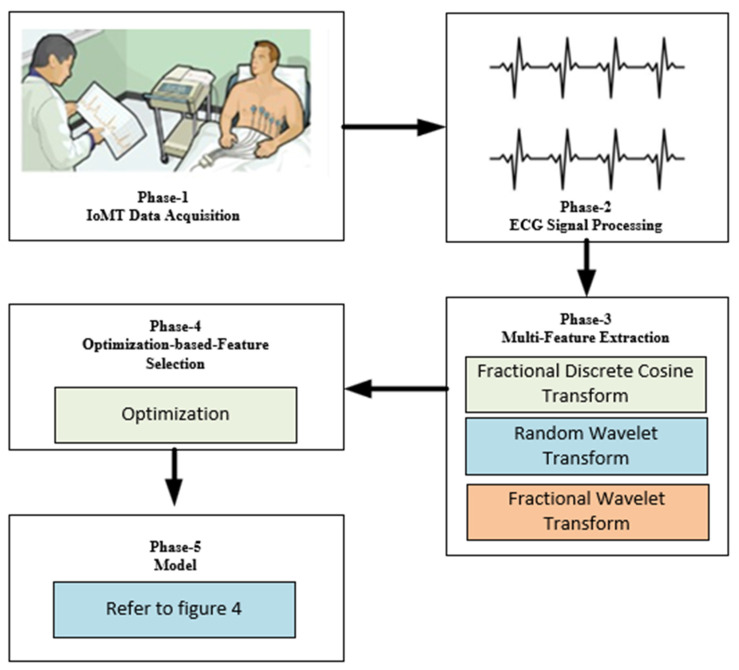
Proposed workflow.

**Figure 2 diagnostics-14-01344-f002:**
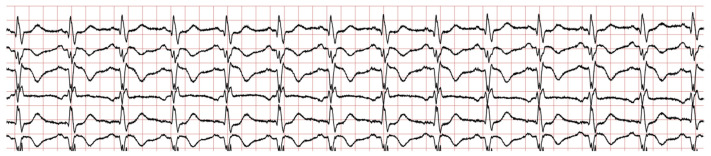
Sample raw signal.

**Figure 3 diagnostics-14-01344-f003:**
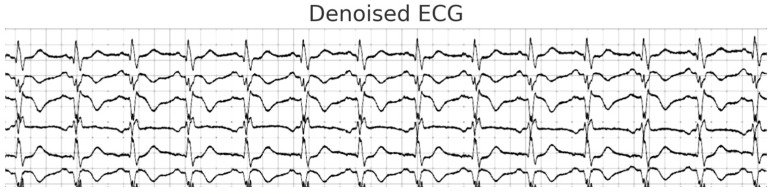
Denoising and segmentation results.

**Figure 4 diagnostics-14-01344-f004:**
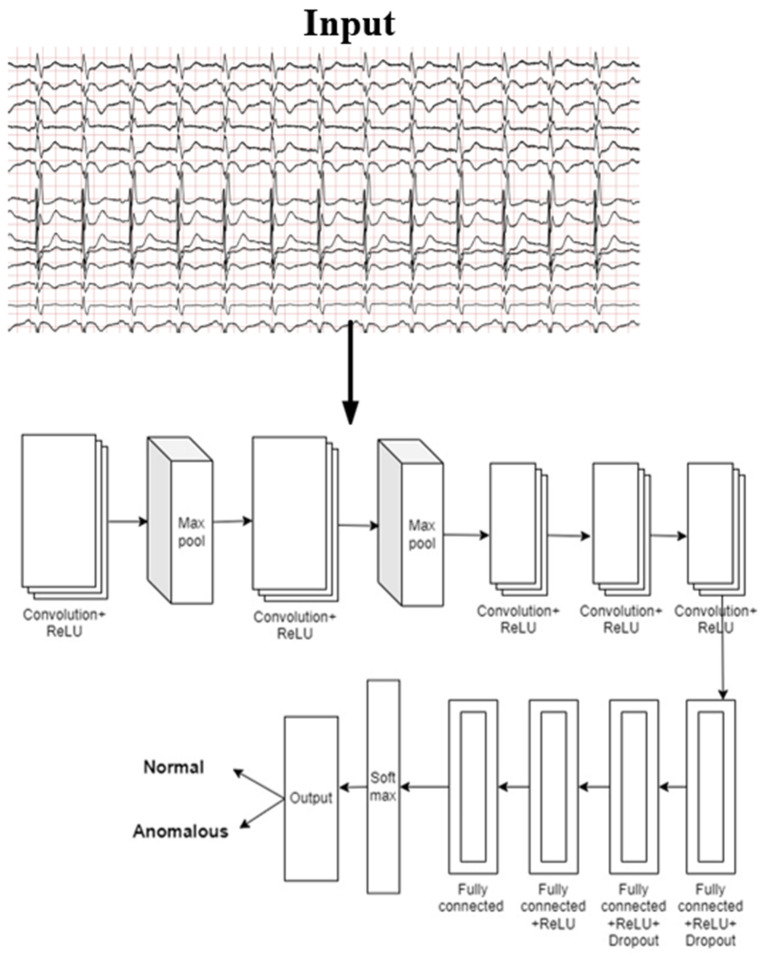
i-AlexNet architecture.

**Figure 5 diagnostics-14-01344-f005:**
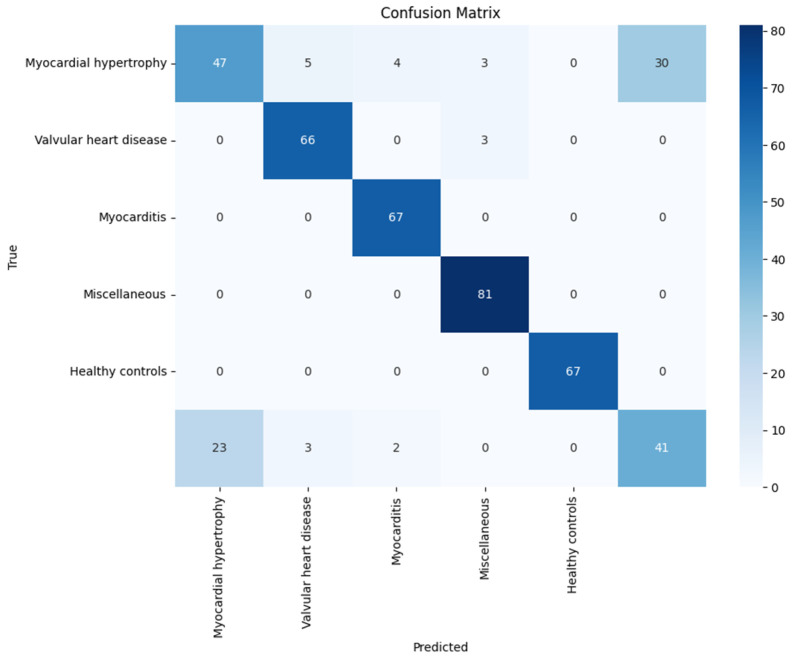
Confusion matrix for classification model performance.

**Figure 6 diagnostics-14-01344-f006:**
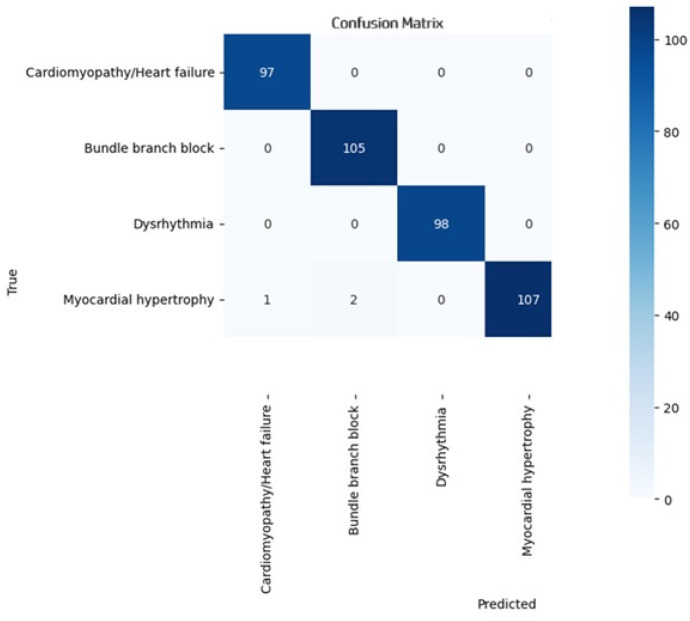
Confusion matrix for classification model performance.

**Figure 7 diagnostics-14-01344-f007:**
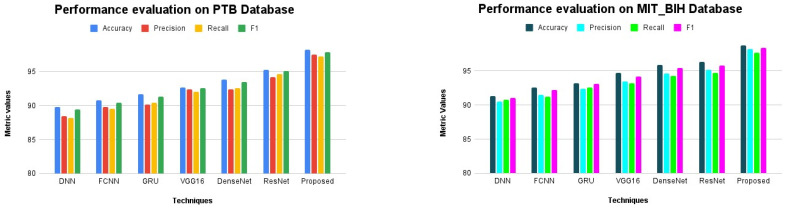
Performance evaluation of proposed system.

**Figure 8 diagnostics-14-01344-f008:**
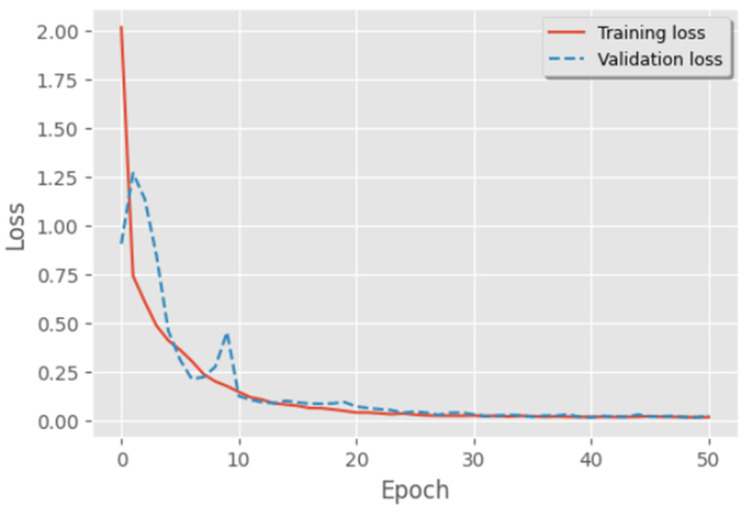
Training vs. validation loss.

**Figure 9 diagnostics-14-01344-f009:**
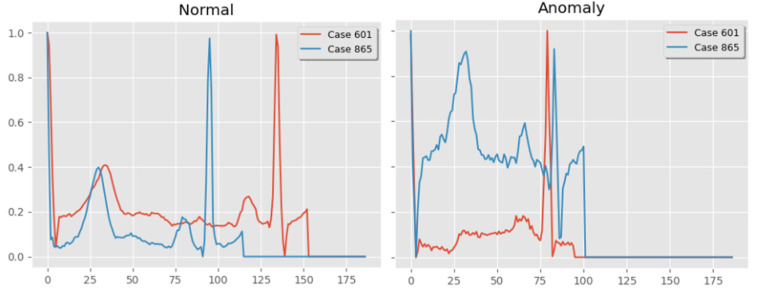
Sample normal and anomalous ECG data.

**Figure 10 diagnostics-14-01344-f010:**
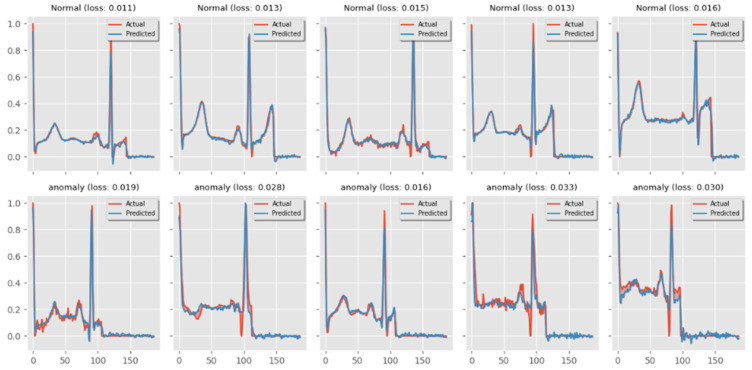
Experimental results of normal vs. anomalous classification.

**Figure 11 diagnostics-14-01344-f011:**
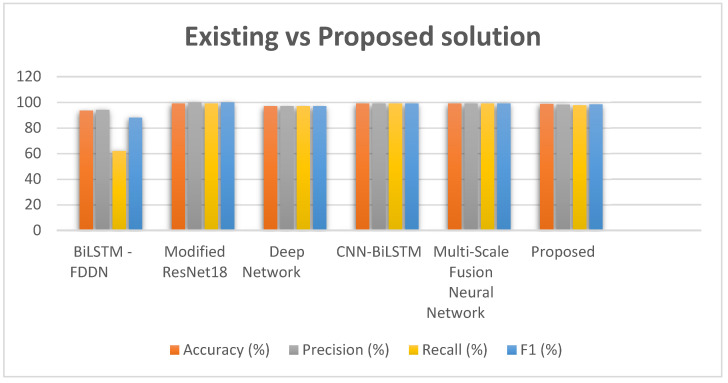
Performance comparison of existing vs. proposed methods [23,25,26,28,30].

**Table 1 diagnostics-14-01344-t001:** Comparison of related works in the literature.

References	Techniques	Dataset Used	Performance
[11]	Deep Neural Networks	MIT-BIH database	Accuracy = 82.3%
			Precision = 81.6%
			Recall = 81.9%
[13]	Support Vector Machine	MIT-BIH normal sinus rhythm database	Accuracy = 84.9%
			Precision = 83.4%
			Recall = 84.5%
[14]	Logistic Regression	PTB database	Accuracy = 83.7%
			Precision = 82.8%
			Recall = 83.4%
[15]	Random Forests	MIT-BIH atrial fibrillation database	Accuracy = 86.6%
			Precision = 85.7%
			Recall = 86.1%
[17]	Sparse Autoencoders	European ST-T database	Accuracy = 92.3%
			Precision = 91.4%
			Recall = 91.8%
[18]	Bidirectional Long Short-Term Memory Networks	MIT-BIH normal sinus rhythm database	Accuracy = 93.6%
			Precision = 93.1%
			Recall = 92.8%
[20]	One-dimensional Convolutional Neural Networks	MIT-BIH atrial fibrillation database	Accuracy = 94.7%
			Precision = 93.5%
			Recall = 94.3%
[21]	Generative Adversarial Networks	European ST-T database	Accuracy = 95.7%
			Precision = 94.2%
			Recall = 95.2%

**Table 2 diagnostics-14-01344-t002:** Distribution of records among diagnostic superclasses in the PTB-XL dataset.

Superclass	Description	Number of Records	Percentage of Total
NORM	Normal ECG	9514	43.57%
MI	Myocardial Infarction	5469	25.06%
STTC	ST/T Change	5235	23.97%
CD	Conduction Disturbance	4898	22.43%
HYP	Hypertrophy	2649	12.13%

**Table 3 diagnostics-14-01344-t003:** Comparison of conventional algorithms.

Techniques	PTB Database				MIT_BIH Database			
	Accuracy (%)	Precision (%)	Recall (%)	F1 (%)	Accuracy (%)	Precision (%)	Recall (%)	F1 (%)
DNN	89.8	88.5	88.2	89.4	91.3	90.5	90.8	91.1
FCNN	90.8	89.8	89.5	90.4	92.6	91.5	91.2	92.2
GRU	91.7	90.2	90.4	91.3	93.2	92.4	92.6	93.1
VGG16	92.7	92.4	92.0	92.6	94.7	93.5	93.2	94.2
DenseNet	93.8	92.4	92.6	93.5	95.9	94.6	94.3	95.4
ResNet	95.3	94.2	94.6	95.1	96.3	95.2	94.7	95.8
Proposed	98.2	97.5	97.2	97.9	98.8	98.2	97.7	98.4

**Table 4 diagnostics-14-01344-t004:** Comparison of existing vs. proposed techniques.

Techniques	Accuracy (%)	Precision (%)	Recall (%)	F1 (%)
BiLSTM -FDDN [23]	93	94	62	88
Modified ResNet18 [25]	99	1	99	1
Deep Network [26]	97	97	97	97
CNN-BiLSTM [28]	99	99	99	99
Multi-Scale Fusion Neural Network [30]	99	99	99	99
Proposed	98.8	98.2	97.7	98.4

## Data Availability

PTB database is available at https://www.physionet.org/content/ptbdb/1.0.0 (accessed on 14 November 2023).

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
