# Peer review of "AI-Driven Real-Time Classification of ECG Signals for Cardiac Monitoring Using i-AlexNet Architecture"

_diagnostics, 2024, doi:10.3390/diagnostics14131344_

Round 1

Reviewer 1 Report

Comments and Suggestions for Authors

The authors proposed an IoMT system driven by AI for cardiac monitoring using the classification of ECG signals. The paper is interesting, but I have the following concerns:

Comment 1. I appreciate the authors' work and would like to suggest a more detailed description of the dataset, particularly whether it is balanced or unbalanced. The authors' mention of determining abnormalities is commendable, but further clarification on the types of abnormalities would greatly enhance the paper. 

Comment 2. The classification proposed is evaluated on an intra or inter-class. This is important since it is associated with the metrics that have been implemented. The evaluation matrix should be shown to determine how good or in what type of instances the model fails. 

Comment 3. The IoMT model is not deeply discussed. It is unclear if the authors proposed an architecture for the IoMT that assures a real-time scenario or if they are implementing this module from somewhere else; hence, they should include some results in this sense. If this is not the case, the authors should modify this contribution to clarify exactly where they are proposing a novelty that impacts the IoMT. 

Comment 4. The paper mainly discusses the classification of ECGs. Nevertheless, the preprocessing stage is crucial to solving the ECG classification; in this sense, the authors should include some results about how to optimize the ECG preprocessing and the multi-feature extraction along with the optimization. Why does the wavelet transform improve other techniques? Which are the final configurations? How do they calibrate each stage? How did they design the experiments? What is the confidence level of their results?

Comment 5. The authors should improve the discussion of the results since they only explain the percentages. They should focus on a deep comparison of the works and algorithms that have been presented in Table 2. 

Comment 6. The authors should improve the discussion by presenting results from all the stages that help to sustain the sentences described in lines 428 to 431.

Author Response

We would like to thank the reviewers for their insightful comments. As a result, our updated and revised manuscript has been significantly enhanced.

Reviewer 2 Report

Comments and Suggestions for Authors

This is a study that proposes a ML system for interpreting an ECG with very high accuracy (greater than 98%) in which the following modifications should be made:

The bibliography should be expanded.

In Fig. 1, Phase 5 is not read, it should be enlarged.

In the results, an example of an interpreted ECG can be included.

In the text, the equations are off-center, as are the links to the results and discussion.

In Fig. 9, include the year in which the previous techniques shown in the image were validated.

It is highly recommended to separate Results and Discussion, since in the discussion the methodology used by the authors should be contrasted with that used in previous works.

With these modifications it can be accepted for publication.

Author Response

(The authors gave the same response as above.)

Round 2

Reviewer 1 Report

Comments and Suggestions for Authors

The authors answered each comment provided in the response letter. Nevertheless, I must only accept the publication of the paper once the authors include each answer inside the paper since they only extend the discussion and the state of the art.

Also, I'd like to know how the authors face the unbalanced data problem since they even generated a binary classification at the beginning of their process. In this sense, how well did the confusion matrix present the results, and then how well did the classification algorithm determine the type of problem detected? In this case, are the results presented from the first classification or the identification of the type of problem detected once they consider an anomalous condition?

Besides, the authors do not explain the IoMT model in the paper. In that case, the paper's name must be changed to highlight the main contributions, as the authors indicated in the Introduction section. 

Author Response

Greetings,

We have updated the entire MS according to the constructive comments provided by the repected reviewer/s. With their constructive comments, our article has improved in its presentation.

with best regards

Round 3

Reviewer 1 Report

Comments and Suggestions for Authors

The authors have improved the paper, making it more fluent. Besides, they answered my questions appropriately. Hence, I suggest publishing the paper.  I only suggest to improve the quality of the images.